

# Patterns of land-cover transitions from satellite imagery of the Brazilian Amazon

Finn Müller-Hansen[1,2], Manoel F. Cardoso[3], Eloi L. Dalla-Nora[3], Jonathan F. Donges[1,4], Jobst Heitzig[1], Jürgen Kurths[1,2], and Kirsten Thonicke[1]

[1]Potsdam Institute for Climate Impact Research, Telegrafenberg A31, 14473 Potsdam, Germany
[2]Department of Physics, Humboldt University Berlin, Newtonstraße 15, 12489 Berlin, Germany
[3]Center for Earth System Science, National Institute for Space Research, Rodovia Presidente Dutra km 40, 12630-000 Cachoeira Paulista, São Paulo, Brazil
[4]Stockholm Resilience Center, Stockholm University, Kräftriket 2B, 114 19 Stockholm, Sweden

*Correspondence to:* mhansen@pik-potsdam.de

**Abstract.** Changes in land-use systems in tropical regions, including deforestation, are a key challenge for global sustainability because of their huge impacts on green-house gas emissions, local climate and biodiversity. However, the dynamics of land-use and land-cover change in regions of frontier expansion such as the Brazilian Amazon is not yet well understood because of the complex interplay of ecological and socio-economic drivers. In this paper, we combine Markov chain analysis and complex

network methods to identify regimes of land-cover dynamics from land-cover maps (TerraClass) derived from high-resolution (30m) satellite imagery. We estimate regional transition probabilities between different land-cover types and use clustering analysis and community detection algorithms on similarity networks to explore patterns of dominant land-cover transitions. We find that land-cover transition probabilities in the Brazilian Amazon are heterogeneous in space and adjacent subregions tend to be assigned to the same clusters. When focusing on transitions from single land-cover types, we uncover patterns

that reflect major regional differences in land-cover dynamics. Our method is able to summarize regional patterns and thus complements studies performed at the local scale.

## 1 Introduction

Land-use/cover change does not only affect local ecosystems and climate but has global consequences for the Earth system (Foley et al., 2005). Land use emits about 25 % of annual greenhouse gases to the atmosphere world wide. Particularly in

tropical regions, increasing demand for food, fibre and biofuels drives land conversion from forest biomes to agriculturally used areas (Lambin and Meyfroidt, 2011). In order to analyze the causes of tropical deforestation, it is thus crucial to understand the dynamics of land-cover changes that occur after deforestation, compare them between regions, and connect them to socio-economic and political drivers. Furthermore, this could help to better understand the effects of land-use intensification that can potentially reverse deforestation trends, as hypothesised in forest transition theory (Meyfroidt and Lambin, 2011).

The Brazilian Amazon is one of the world's key regions with highly dynamic land-use change and is subject to multiple pressures (Laurance and Williamson, 2001; Keller et al., 2009; Davidson et al., 2012). Economic activities such as unsustain-


able logging and agricultural expansion of cattle ranching and soy bean cultivation lead to a fragmentation of the landscape resulting in biodiversity loss (Laurance et al., 2002). Global climate change may decrease precipitation and increase forest fires (Chen et al., 2011). All these pressures are increasing the risk of destabilizing the ecosystem and crossing a tipping point with irreversible consequences (Lenton et al., 2008; Nepstad et al., 2008; Staal et al., 2015).

In the 1970s and 80s, deforestation was mostly driven by large infrastructure and settlement programs, but more recent years saw mainly market drivers pushing the deforestation frontier further, while government programs tried to contain it (Fearnside, 2005). Since 2005, deforestation rates in the Brazilian Amazon have been reduced enormously. In recent years, the rates are fluctuating between 5000 and 6000 $km^2$ per year, which is a reduction of about 80% compared to the peak of deforestation activities in 2004 (INPE, 2016). The changes are explained by new monitoring programs, public policies and supply chain

interventions (Nepstad et al., 2014; Dalla-Nora et al., 2014; Gibbs et al., 2015). However, there are warnings that deforestation may increase again (Fearnside, 2015; Aguiar et al., 2016).

In order to understand deforestation rates, it is crucial to take subsequent land-uses and their dynamics into account. This paper focuses on developing methods to detect patterns of land-cover dynamics using data from remote sensing and identifying large-scale differences between subregions of the Brazilian Amazon as a sample region. To do so, we draw on the theory of

Markov chains that has been used in the context of land-system science to describe and analyze land-cover dynamics (Bell and Hinojosa, 1977; Baker, 1989). Markov chains are stochastic systems that are described by transition probabilities between discrete states, here referring to a specific land-use or land-cover type. An ensemble of such chains describes a collection of land patches that undergo stochastic transitions between land-cover classes. Because simple Markov models do not take spatial correlations into account, they often form only one part of hybrid land-cover models that introduce stochasticity into the

model (see e.g. Brown et al., 2000; Subedi et al., 2013). For example, Fearnside (1996) applied a Markov analysis to estimate greenhouse gas emissions from land-use change in the Brazilian Amazon and found that carbon storage in the land system decreases as it approaches an equilibrium.

In the past, most studies using Markov analysis focused on small regions due to limited data availability. Modern geographic information systems (GIS) enable the detection of land-cover changes at an unprecedented scale using satellite images (Lu

et al., 2004). Automated algorithms allow the classification of land use and land cover of vast regions. Furthermore, it is possible to compare the land-use dynamics between different subregions and find differences and similarities based on consistent datasets. For example, Levers et al. (2015) combined different sources of land-use indicators and used self-organizing maps to identify archetypical land uses and regions with similar land-use change in Europe.

In this study, we use Markov transition probability matrices as a descriptor of aggregate land-cover dynamics estimated from

high-resolution land-cover data for 3 time slices of land-cover over 6 years in the Brazilian Amazon. To our knowledge, Markov analysis has so far not been applied to investigate interregional heterogeneity of land-cover dynamics. This paper explores this idea by comparing transition matrices from different subregions in the Brazilian Amazon to identify patterns of similar land-cover dynamics drawing on large data sets derived from satellite imagery. While previous studies mostly worked with predefined regions to compare land-cover dynamics, we develop methods to identify regions with similar land-cover dynamics

which allows a large-scale analysis of land-cover change patterns. With this methodology we approach the hypothesis that





different land-cover dynamics can be identified by the characteristics of their transition matrix and a partition of subregions, for example into remote, frontier and consolidated areas, can be detected from the data.

The paper is structured as follows: In the subsequent Sections 2 and 3, we present the details of the proposed method and describe the data that we apply it to. Section 4 gives results from the analysis and discusses them, pointing to possible interpretations but also restrictions of the method. Section 5 concludes with an outlook on how the method could be applied to further analyses.

## 2 Data

In this study, we use land-cover maps of the Brazilian legal Amazon (cp. Fig. 1) produced by the TerraClass project (INPE and EMBRAPA) for the years 2008, 2010 and 2012. The land-cover maps are derived from high-resolution Landsat-5 thematic mapper (TM) and MODIS imagery using a mix of techniques including supervised learning and classification by spectral properties of different land-cover types and their annual variations (for details, see Almeida et al., 2016; Coutinho et al., 2013). The maps consist of polygons that represent patches of land attributed to one of 16 specific land-cover types (see Table S1 in the supplementary materials). The maps are based on the PRODES project that distinguishes between forest, patches not belonging to the rain forest biome (mainly savanna), hydrography (i.e. lakes and rivers), and deforested patches larger than 6.25 ha (INPE, 2016). TerraClass further specifies the land-cover of formerly deforested areas according to 12 types including different kinds of pasture land, secondary vegetation and annual crops. Coutinho et al. (2013) evaluated the accuracy of land-cover detection using the method described in Congalton and Green (2009). Considering a very small sample of the data set, they found up to 58 % commission and up to 34 % omission errors. Almeida et al. (2016) found that the dominant land cover on previously deforested land is pasture (62% as of 2008) followed by secondary vegetation (21%). Annual crops only covered about 5% of the total deforested areas.

This paper focuses on relevant transitions between major land-cover classes occurring in different subregions of the Brazilian Amazon. Therefore, we first exclude patches that could not be classified, e.g. due to cloud cover. Second, we discard land-cover types that do not change by definition, i.e. lakes and rivers and patches not belonging to the rain forest biome. Third, we aggregate similar land-cover types into six new classes. These classes combine different types of less intensively used pasture as well as types that only make up small fractions of the Amazon like mining and urban patches (see Table S1) and group land-cover types between which high confusion errors exist, thus decreasing them. In a final step of the data preparation, we assign patches to $N$ different subregions. Depending on the scale of spatial aggregation of our analysis, the subregions either correspond to the legal municipalities of the Brazilian Amazon ($N$=770, as of 2007) or to the mesoregions ($N = 30$) as defined by the Instituto Brasileiro de Geografia e Estatística (Brazilian Institute of Geography and Statistics, IBGE (2016)).




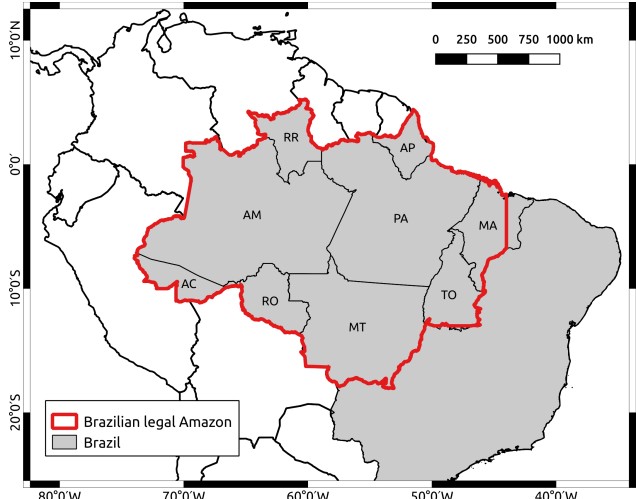

**Figure 1.** Map of the Brazilian legal Amazon and its nine federal states: Acre (AC), Amapá (AP), Amazonas (AM), Maranhão (MA), Mato Grosso (MT), Pará (PA), Rondônia (RO), Roraima (RR) and Tocantins (TO).

## 3    Method

In order to compare land-cover dynamics between different subregions of the Amazon, we proceed in two steps: First, we calculate the area in a given region that undergoes a transition from one land-cover type to another between two reference years (including the lumping of several land-cover types into one class) and normalize the obtained matrices. Second, we

5   compare the transition matrices between subregions by means of cluster analysis and network methods. In this section, we describe the steps of the method in detail.

### 3.1    Transition matrices

Markov chains are stochastic systems, in which the probability distribution of the next time step only depends on the current state of the system, i.e. the system has no memory. A subregion can be thought of as consisting of a number of land patches that

10  undergo transitions between land-cover classes. Markov analysis then describes how the set of patches may change over time. Although the Markov property, i.e. that the transition probability only depends on the present state of the system, can be shown to hold approximately for land-use systems (Robinson, 1978), the transition rates are generally not constant over time, which means the system is not stationary. This is not surprising because of the various climatological and socio-economic drivers and political decisions influencing land-cover dynamics (Walker, 2004). Even though Markov chain analysis may oversimplify

15  land-cover dynamics because it does not take the underlying processes explicitly into account and may therefore not be suitable to project future land-cover change, it serves here as a first approximation in obtaining a general understanding of the land-cover dynamics observed in the data.





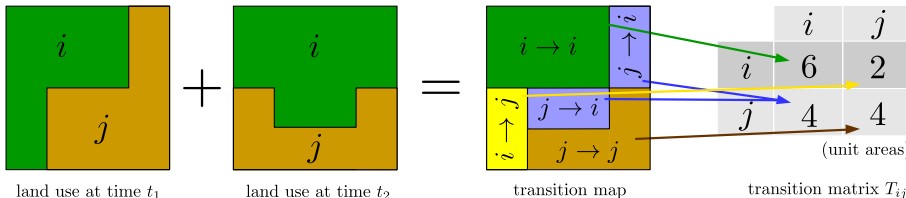

**Figure 2.** Illustration of the geometric union operation that combines the information of two land-cover maps into a transition map and how the transition matrices are obtained from this map.

We obtain the transition matrices of subregions by calculating the areas in a given subregion that undergo a transition from a land-cover class $i$ to another class $j$. The transition matrix of one subregion $\mathbf{T}(t)$ is an $n \times n$ matrix with elements $T_{ij}(t)$, $i, j \in \{1, ..., n\}$, where $n$ is the number of land-cover classes. The transition matrix depends on time, indicating the non-stationarity of the Markov process. In the following, however, we omit the time dependence for ease of notation. With the

aggregation described above, $n = 6$. We estimate $T_{ij}$ from the data by first projecting the coordinates of the patches (in the data given in the *South American Datum (SAD69)* coordinate system) to the *South America Albers Equal Area Conic* projection. Second, we compute the geometric union with a GIS software combining the information contained in the two land-cover maps of the reference years into one data set. Finally, we sum up the area of all patches in one subregion that undergo the same transition. Figure 2 illustrates the creation of the transition matrix $T_{ij}$ from the data.

To estimate transition probabilities, we have to normalize the transition matrices. Thereby, we also make subregions of different total area comparable. We normalize the rows of the transition matrices to 1, which allows us to focus on relative changes in single land-cover classes,

$$p_{ij} = \frac{T_{ij}}{\sum_k T_{ik}} \text{ for } i, j : 1...n. \tag{1}$$

The normalization does not work if one land-cover class $i$ does not figure in the data of one subregion as $\sum_k T_{ik}$ would be
equal to zero. In such cases, we set the diagonal element $T_{ii} = 1$, implying that we handle the land-cover class in the particular subregion as if no change occurs.

In statistical terms, $\mathbf{p} = (p_{ij})$ is a stochastic matrix (compare Norris, 1997) with the properties $p_{ij} \geq 0$ and $\sum_j p_{ij} = 1$ for $i = 1...n$. It corresponds to the maximum likelihood estimation of the transition probability matrix of a first order Markov chain where land-cover classes correspond to the states of the Markov chain and the rows of $\mathbf{p}$ specify the transition probabilities
between the states (Anderson and Goodman, 1957).

Figure 3(a) presents a visualization of the Markov chain and the calculated transition probabilities estimated for the whole Brazilian Amazon. The figure shows that there are transitions between almost all aggregated classes, but they occur with very different probabilities. After deforestation, about two thirds of the areas are used as pasture, whereas the rest is mostly classified as secondary vegetation. Furthermore, transitions occur frequently between pasture partly covered with woody vegetation (dirty
pasture) and clean pasture. The former makes also frequent transitions to secondary vegetation. Finally, there are considerable





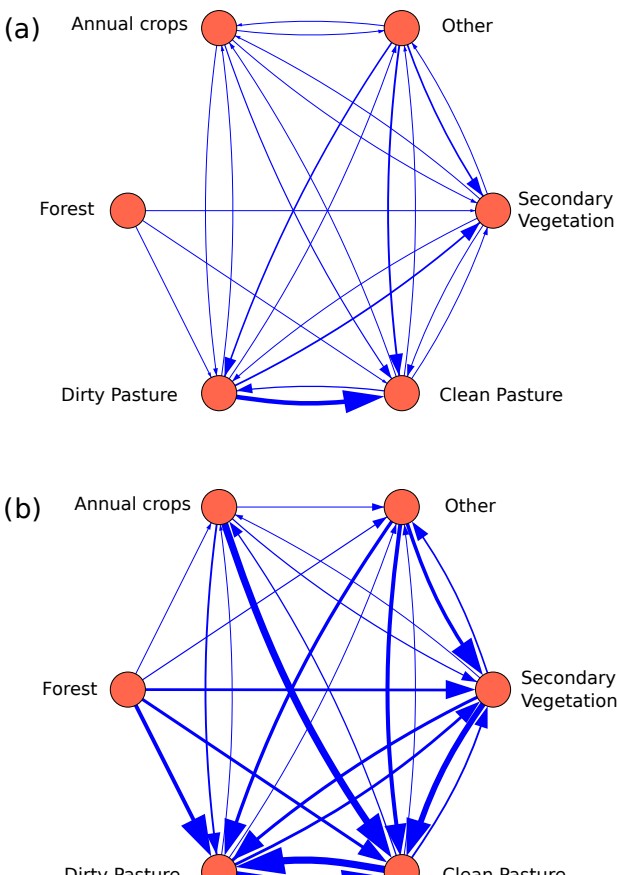

**Figure 3.** Illustration of the normalized transition matrices between simplified classes derived for the whole Brazilian Amazon from the TerraClass data set (changes between 2010 and 2012): (a) Markov transition matrix **p** (self-loops omitted) (b) conditional transition matrix **q**. The strengths of the arrows are scaled with the transition probabilities except for those representing small values. Arrows with very small values (below 0.005) are not shown. The values are given in Tables S2 and S3.

transitions from and to the "other" class, in which we aggregated the minor land-cover types mosaic of uses, urban area, mining, others and reforestation from the original TerraClass classification.

Alternatively to the Markov analysis, one could normalize the sum of the transition matrix elements $T_{ij}$ to one. Such a normalization would keep the information on the initial distribution of land-cover classes in one subregion but would not allow to analyze relative changes in individual land-cover classes.

The transition probability matrix **p**, representing the dynamics of an underlying Markov chain process, includes information on the patches that undergo changes and the patches that remain in their land-cover class. To only consider changes, we set the



diagonal elements to zero before normalizing the rows of $\mathbf{T}$ to 1,

$$
q_{ij} =
\begin{cases}
\frac{T_{ij}}{\sum_{k \neq i} T_{ik}} & \text{for } i \neq j \\
& \text{for } i = j.
\end{cases}
\tag{2}
$$

$\mathbf{q} = (q_{ij})$ thus estimates the probability to make a transition from a single land-cover class $i$ conditional on that there is a transition to a different land-cover class $j$. Figure 3(b) shows a visualization of this conditional transition matrix for the whole

Brazilian Amazon. For land-use classes that have a high proportion of patches remaining in the same class, this figure allows inspecting the relative shares of transitioning patches more easily.

The normalized matrices $\mathbf{p}$ and $\mathbf{q}$ describe the transitions between all land-cover classes. In the following we are particularly interested in comparing transition probabilities from a single land-cover class to all others, formally represented by the rows of the normalized matrices. If we only focus on the rows, we solve the above-mentioned problem of missing land-cover classes

in a subregion by simply discarding the respective subregions from the analysis. To increase the robustness, we also discard subregions having less than $1\mathrm{km}^2$ of the considered land-cover class.

As described above, we estimated the normalized transition matrices $\mathbf{p}$ and $\mathbf{q}$ for all mesoregions and municipalities separately. In general, the lower the spatial aggregation, the higher is the variability in space and in time, which we observe when comparing the mesoregion and municipality maps and transitions between different times. Figure 4 shows two exemplary com-

ponents of the matrices $\mathbf{q}$ calculated for each municipality. The two maps highlight these subregions in darker colors in which the transition probability from clean pasture to secondary vegetation and vice versa is high compared to transitions to other land covers. In Fig. 4(a), we can observe that transitions from clean pasture to secondary vegetation are infrequent compared to other transitions except in the central North and the South West. Figure 4(b) suggests that along a horizontal band from the West to the East and in the North (state of Roraima) the transition probability from secondary vegetation to clean pasture is

higher than in the other parts of the Brazilian Amazon. The maps in Fig. 4 and similar maps for all other possible transitions contain the information that we aim to aggregate using clustering analysis. The next section therefore describes this second step of our method.

## 3.2 Clustering analysis

Clustering methods are a basic techniques described in the machine learning and data mining literature (Jain and Dubes, 1988;

Gan et al., 2007). In recent years, the basic problem of clustering nodes in complex networks has also gained a lot of interest in complex systems science (Fortunato, 2010). In this paper we choose a combination of established and more recent clustering methods to compare and test the robustness of our results. The chosen established methods are hierarchical clustering and the k-means algorithm. The other methods are based on complex networks that we construct from a difference measure. To partition the network, we apply two different community detection algorithms, the fastgreedy and Louvain algorithms (Clauset

et al., 2004; Blondel et al., 2008).

The first method applies hierarchical clustering that merges data points or clusters based on their distance in the abstract data space. In the context of this analysis, a data point $x$ is either a full normalized transition matrix (flattened, such that $x \in \mathbb{R}^{n^2}$)





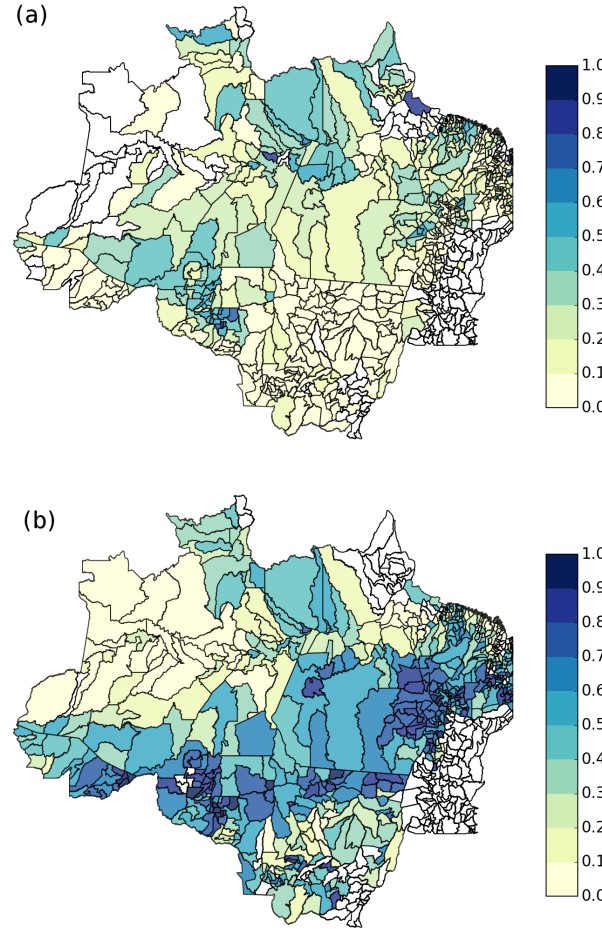

**Figure 4.** Map of two selected components of the conditional transition matrices **q** for each municipality of the Brazilian legal Amazon. Colors indicate the shares of areas that make a transition from (a) clean pasture to secondary vegetation and (b) secondary vegetation to clean pasture.

or a single row of such a matrix ($\boldsymbol{x} \in \mathbb{R}^n$). Each data point corresponds to an individual subregion. We choose to calculate the distance between two data points $\boldsymbol{x}$ and $\boldsymbol{y}$ by the $\ell_1$ norm, also called Manhattan distance, $d(\boldsymbol{x}, \boldsymbol{y}) = \sum_i \mathrm{abs}(x_i - y_i)$. This distance is easy to interpret in the context of probabilities and compared to the euclidean metric does not punish outliers of a cluster as much. The distances between two clusters or one cluster and one data point are calculated using the complete linkage

5  algorithm that takes the maximal distance between the points of two clusters. This algorithm identifies compact clusters with small diameters (Jain and Dubes, 1988). Hierarchical clustering produces a dendrogram of cluster partitions. The clusters are obtained by cutting the dendrogram at a certain level determining the number of clusters.





The second method uses the k-means algorithm. The algorithm works in an iterative manner: It associates data points to centroids and adjusts the position of the centroids by minimizing the within-cluster sum of squared distances. The k-means algorithm inherently requires the choice of the euclidean metric to calculate distances.

The network methods both require the construction of a similarity network first. In the network, each node $v_\alpha$ represents a subregion and nodes with similar dynamics are linked by an edge $e_{\alpha\beta}$, where the greek character indices refer to subregions. The connectivity of the network can also be represented by an adjacency matrix $\mathbf{A} = (A_{\alpha\beta})$. To determine the similarity, we use a normalized version of the Manhattan distance as the difference measure $d(\boldsymbol{x}, \boldsymbol{y}) = \frac{1}{2k} \sum_i \mathrm{abs}(x_i - y_i)$, where $k$ is the number of land-cover classes $n$ if we compare whole transition matrices and $k = 1$ if we only consider transitions from single land-cover classes. The metric is zero if and only if transition probabilities are equal and $1$ if they are completely different. We set a threshold $d_{th}$ to transform the data into a network with the adjacency matrix $\mathbf{A}$:

$$
A_{\alpha\beta} = \begin{cases} 1 & \text{if} \quad d(\boldsymbol{x}_\alpha, \boldsymbol{x}_\beta) < d_{th} \\ 0 & \text{else.} \end{cases} \tag{3}
$$

This adjacency matrix contains all information on the similarity network. The threshold $d_{th}$, which determines the subregions that are connected, is chosen such that only links that are significantly different from a distribution of difference measures of random vectors or matrices are realized. In order to obtain $d_{th}$, we apply Monte Carlo simulation: we generate a large number ($10^6$) of random samples of vectors or matrices, the values of which are drawn from a uniform distribution and rows are normalized. From the computed distribution of pairwise difference measures, we use the 5th percentile to determine the threshold $d_{th}$.

A visualization of such a similarity network is shown in Fig. 5 for transitions from clean pasture to other land-cover types. The nodes of the network represent data points for the municipality drawn around it. Links are drawn between regions that have a difference measure below the significant threshold $d_{th} = 0.11$, which we obtain as described above from a Monte Carlo calculation of normalized random vectors of dimension $4$ (because transitions to 4 other classes are possible). A visual inspection of the network suggests that similar transition probabilities are detected in regions of the Eastern and the Southern Amazon, whereas there are less similar transitions in the Northern part. The inset in Fig. 5 furthermore shows a histogram of all pairwise differences. The threshold is indicated as a red vertical line. From tests with different thresholds and different underlying data, we can conclude that the patterns observed in the similarity networks hardly depend on the exact choice of the threshold (or link density). Thus the construction of the network is robust with respect to variations of the threshold.

The visual inspection of similarity networks is difficult and may not be reliable. Therefore, we applied community detection algorithms to the networks to infer information about the network structure. These algorithms identify clusters of nodes on the network (in the literature the clusters are often called communities, hence the name) that have a high internal connectivity. Most of these algorithms are based on the idea of optimizing modularity $Q$, a network measure that compares the frequency of





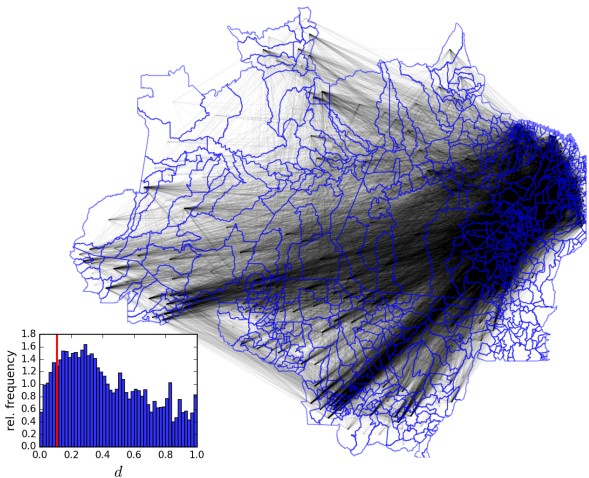

**Figure 5.** Illustration of a similarity network with a spatial division in municipalities for transitions from clean pasture to other land-cover classes between 2010 and 2012. Inset: Histogram of difference metric values with threshold in red.

links inside of communities to the frequency of links between communities (Fortunato, 2010). For a network with adjacency matrix $\mathbf{A}$ and clusters $C$, the modularity is given by

$$Q = \frac{1}{2m} \sum_{\alpha,\beta} A_{\alpha\beta} - \frac{k_\alpha k_\beta}{2m} \delta(C_\alpha, C_\beta), \qquad (4)$$

where $k_\alpha = \sum_\beta A_{\alpha\beta}$ is the degree of node $\alpha$ and $m$ is the number of edges in the network. The term $\delta(C_\alpha, C_\beta)$ only gives

5   a contribution if nodes $\alpha$ and $\beta$ belong to the same cluster. In the following, we constrain our comparison to the fastgreedy and the Louvain algorithms, which are computationally efficient and yield comparatively high modularity values. The general idea of the fastgreedy algorithm as described in Clauset et al. (2004) is to subsequently join clusters such that the increase in modularity is highest after the join. This produces a dendrogram, similar to the output of the hierarchical clustering method, which can be cut at the level of highest modularity $Q$. In contrast, the Louvain algorithm developed in Blondel et al. (2008)

10   proceeds in two iterative steps: It first checks subsequently if the reassignment of single nodes to other clusters leads to an improvement in modularity. In a second step, it builds a new network combining all nodes of a community found in the previous step into one node and sums up all edges between communities to form weighted new edges.

In the following, we apply these algorithms to the same heterogeneous data. A comparison between the different methods will show whether the clustering can be considered robust.





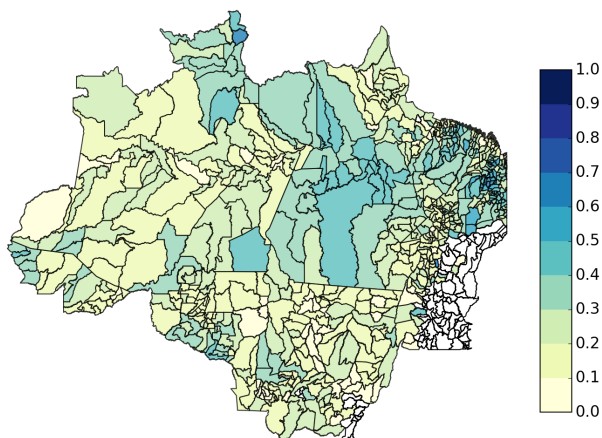

**Figure 6.** Relative areas that undergo changes in land-use classes between the years 2010 and 2012 (excluding primary forest).

## 4 Results and Discussion

This section describes patterns of land-cover change found in the Brazilian Amazon when applying the clustering algorithms of differently normalized transition matrices or single rows of them. We show the spatial comparison of transitions between 2010 and 2012 with the threshold for the construction of the similarity networks set to $d_{th} = 0.11$ (see Section 3.2). Comparisons of

transitions between other years are shown in the supplementary material.

As explained in the methods section, we considered different normalizations of the transition matrices: the Markov matrices **p** that also contain information about patches remaining in the same land-cover class and conditional transition matrices **q** that disregard this information. First, we note that the majority of land patches does not change its class from one time step to the next. This is illustrated in Fig. 6, where the relative area of patches that make a transition to a different land-cover class

is plotted (excluding primary forest), i.e. the sum of the diagonal elements of the transition matrix divided by the sum of all elements. Only in the Central Amazon and in some of the smaller municipalities there are considerable fractions of up to 50% of the area undergoing a change in land-cover class. Because we are interested in the changes, we will focus our discussion first on the conditional transitions matrices **q** and compare only single rows between the municipalities.

As an example, Fig. 7 displays the result of the clustering analysis for transitions from clean pasture to other land-cover

classes. To make the clustering comparable we fixed the number of clusters for the hierarchical and k-means clustering to the one obtained from the fastgreedy network clustering algorithm. As we can see from the figure, there are clearly distinguishable clusters in the South and the North West of the Amazon colored in orange and cyan for all four different clustering algorithms. These clusters are identified independently of the chosen clustering algorithm. In the other parts of the Amazon region, the clusters vary dependent on the applied clustering algorithm. Both network community detection algorithms identify similar

clusters, even though the Louvain algorithm finds seven and the fastgreedy algorithm reveals five communities in the data.



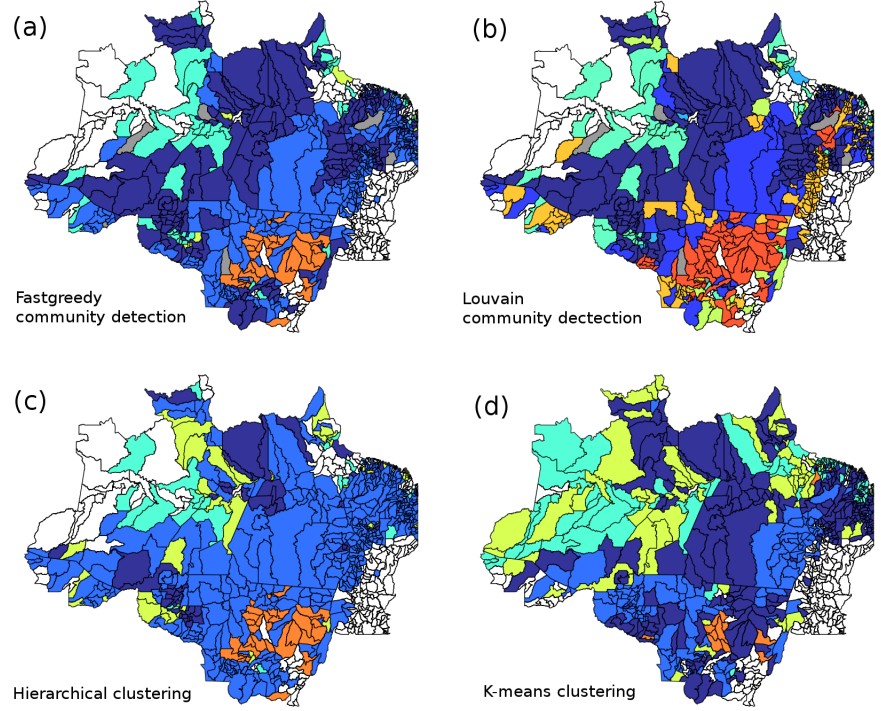

**Figure 7.** Comparison of network (a, b) and classical (c, d) clustering algorithms for conditional transitions from clean pasture to other land-cover classes between 2010 and 2012. Each cluster is visualized by one color. White regions lack data to estimate the transition matrix, grey regions are not connected to the similarity network. The number of clusters for the hierarchical and k-means clusters was chosen to match the outcome of the fastgreedy algorithm (5). The Louvain algorithm detects 7 clusters.

Also, some clustering algorithms seem to find two clusters for a group of municipalities, where other algorithms only find one (compare e.g. the fastgreedy with the k-means algorithm). In addition to the two relatively stable clusters, we can observe in Fig. 7 that most clusters consist of adjacent municipalities. This suggests that neighboring municipalities have a high likelihood to exhibit similar relative land-cover changes.

In order to interpret the clusters, we analyzed the cluster centroids, i.e. the mean of all data points in a cluster weighted by the area of the considered land patches in the subregion. Figure 8 shows the cluster centroids from the hierarchical clustering. The bars indicate the shares of patches making a transition from clean pasture to another land-cover class and thus show which transitions are dominating or are absent in the cluster. They allow a straight-forward interpretation of different clusters: For instance, in municipalities belonging to the orange cluster, most of the areas are converted to annual crops while only a small fraction makes the transition to dirty pasture. This is in line with a previous study by Macedo et al. (2012) who found that cropland expanded mostly into pasture in the region between 2006 and 2010. The orange cluster is located inside the Mato Grosso State, one of the biggest producers of soybeans in Brazil, which are detected as annual crops in the data. As we can see,



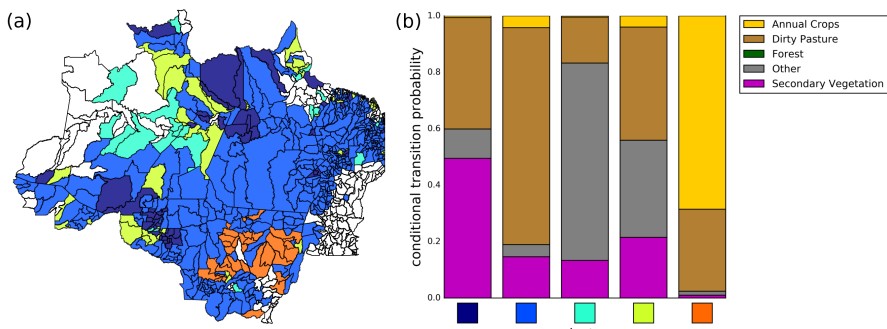

**Figure 8.** (a) Hierarchical clustering with conditionally normalized transition probabilities from clean pasture to other land-cover classes between 2010 and 2012, as in Fig. 7(c). (b) Cluster centroids showing the conditional transition probabilities of the average over the respective cluster indicated by cluster color.

the clusters generally differ by their relative shares of land-cover types such as dirty pasture and secondary vegetation. When comparing the cluster centroids between algorithms, these shares differ for the unstable clusters while the cluster centroids of the stable clusters are almost the same.

So far, we discussed transitions from clean pasture to other land-cover classes as one example. But our analysis has shown that the stable clusters identified in Fig. 8 can also be found when considering transitions from other land-cover classes, e.g. from secondary vegetation (see Figs. S1 and S2). However, the same patterns are not found for all transitions from single land-cover types. This is not surprising considering typical land-cover sequences (often called land-use trajectories) that follow total deforestation and are discussed in the literature (Ramankutty et al., 2007; Alves et al., 2009; de Espindola et al., 2012). According to these studies, a common trajectory is that cleared forest patches are converted to pasture land or used for small-scale subsistence agriculture. After a while, as the soil degrades, the areas are often abandoned leaving them for regrowth of secondary vegetation. Later, they may be cleared again and reused as pasture or they are converted to more intensive agricultural cropland, e.g. for soy bean cultivation. These accounts are generally consistent with our results.

In addition to the clustering based on transitions from single land-cover classes, we tried to identify regions that are similar regarding the transitions between all land-cover classes. The clustering based on the full Markov matrices $\mathbf{p}$ proved to be very unreliable due to the high heterogeneity and dimensionality of the data (see Fig. S3). Furthermore, the analysis of the difference measure showed that only a small fraction of municipalities are significantly similar to each other compared to random matrices. The clustering based on the full conditional transition matrix $\mathbf{q}$ turned out to be highly dependent on the assumptions we made to fill in missing data. Thus, we can conclude that a general classification of land-cover dynamics only based on the full transition probability matrices between different land-cover types is not reliable.

This may have several reasons: First, the underlying processes of land-cover change in the Amazon are very heterogeneous in space and time and are therefore difficult to compare. Second, the areas of the municipalities may be too small for a reliable estimation of transition probabilities. For this reason we also analyzed the transition matrices at the level of mesoregions (see



Fig. S5). However, there was no reliable clustering at this spatial aggregation either. Third, the classification of land-cover types in the TerraClass data set comes with considerable errors. We tried to reduce the errors by aggregating some of the original classes. However, there is not yet an evaluation of the performance of change detection available for this data set, which makes an estimation of the errors in our analysis difficult.

The Brazilian Amazon has been broadly divided into mostly undisturbed, frontier and consolidated areas. For example, Becker (2005) distinguishes between the Arch, i.e. densely populated areas in the South and the East of the legal Amazon, new frontier regions in the Central Amazon and the mostly undisturbed West. Aguiar et al. (2007) used this partition to analyze inter-regional differences in factors potentially determining deforestation and found that the importance and combination of factors such as protected areas, distance to roads and access to markets differs between the three subregions. Although these
studies focus on the 1990s and large-scale socio-economic patterns may have changed since then, our analysis suggests that there are no clear patterns in the estimated transition probabilities which correspond to a spatial partition such as the one proposed by Becker (2005).

## 5  Conclusions

This paper has explored variations of a method that is able to provide important information on the dynamics of land covers,
including the ability to quantify and compare land-cover transition frequencies and identify regions of similar patterns of land-cover change. We have applied different clustering techniques to find patterns in the subregional transition probabilities between land-use classes and detected patterns of subregions presenting similar transitions dynamics that are consistent with other studies. In some regions, such as Northern Mato Grosso where transitions from pasture to annual crops dominate, spatial patterns of relative land-use changes are consistent between different clustering methods. However, our analysis also indicates
that relative land-use changes do not follow clearly distinguishable patterns that are linked to earlier socio-economic partitions of the Brazilian Amazon.

The integration of socio-economic data into the framework described in this paper could potentially yield insights about the underlying drivers and processes of land-cover transitions and how regionally different transition probabilities are determined. Furthermore, the analysis presented in this paper could potentially be used to parametrize models of land-cover change that
track aggregate areas with different land-cover types. By controlling specific transition rates as functions of socio-economic drivers, such models, to be developed in future research, could give rough ideas about possible future developments of land cover and thus support the planning of future land-use policies in the Amazon region.

*Acknowledgements.* F.M.-H. acknowledges funding by the DFG (IRTG 1740/TRP 2011/50151-0). J.F.D is grateful for financial support by the Stordalen Foundation (via the Planetary Boundary Research Network PB.net) and the EarthLeague's EarthDoc program. We thank
Ana Cano Crespo for help with the TerraClass data and Tim Kittel, Catrin Ciemer and Silvana Tiedemann as well as the members of the ECOSTAB and COPAN flagships at PIK for fruitful discussions. The data preparation for this paper was carried out using ArcGIS with a



licence provided by the German Research Centre for Geosciences (GFZ, Potsdam). The data analysis relies on the following python packages: scipy, scikit-learn, pandas, igraph, networkx, shapefile and matplotlib. We thank all the contributors and developers of these packages.



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
