# Peer review of "A matrix clustering method to explore patterns of land-cover transitions in satellite-derived maps of the Brazilian Amazon"

_Nonlinear Processes in Geophysics, 2016_

## Referee Comment (RC1) · A. Tsonis (Referee) · 21 Oct 2016

The authors are experts in the field and their work is very good. I have no problem with this paper being accepted as is.

---

## Referee Comment (RC2) · Anonymous Referee #2 · 26 Dec 2016

The paper focused on an important topic, contains thorough investigations and is clearly written. I would strongly recommend this paper.

Some minor issues that could be amended at the discretion of the authors:

1) The segmentation procedure is an issue. While in terms of management quality monitoring looking at administrative divisions such as municipalities might appear most relevant, the authors nevertheless mention, that this leads to certain limitations, caused e.g. by the area of municipalities being too small for providing reliable estimates of transition probabilities. For a better detection of regional trends, maybe not in the context of particular municipalities, why not use some standard segmentation procedures, e.g. Voronoi diagrams? This could also simplify the normalization procedures.

2) Very minor technical points:

– p.1, l. 14 world wide -> worldwide? – p.11, section title "Results and Discussion" , but there are many original results prior to this section, such as segmentation, network design etc. If not enforced by journal format, maybe use more specific (sub)sectioning?
* * *

---

## Author Comment (AC1) · 12 Jan 2017

We thank the referees for their effort to evaluate our manuscript and their feedback. In the following, we respond to the comments.

**1    Referee #1**

Referee comment: *The authors are experts in the field and their work is very good. I have no problem with this paper being accepted as is.*

Response: We thank the referee for the unconditional approval of our work.

**2 Referee #2**

Referee comment: *The paper focused on an important topic, contains thorough investigations and is clearly written. I would strongly recommend this paper.*

*Some minor issues that could be amended at the discretion of the authors: 1) The segmentation procedure is an issue. While in terms of management quality monitoring looking at administrative divisions such as municipalities might appear most relevant, the authors nevertheless mention, that this leads to certain limitations, caused e.g. by the area of municipalities being too small for providing reliable estimates of transition probabilities. For a better detection of regional trends, maybe not in the context of particular municipalities, why not use some standard segmentation procedures, e.g. Voronoi diagrams? This could also simplify the normalization procedures.*

Response: We chose the segmentation of the region into municipalities for two major reasons: First, the choice of municipalities makes our analysis compatible with other data, e.g. socio-economic data provided by the IGBE or agricultural census data. Second, municipalities are sized – among other criteria such as the physical geography – according to their population. Assuming that the land-use activities are proportional to the population in the area (leaving urban population aside) thus makes it reasonable to look at this kind of spatial segmentation. The alternative segmentation into grid cells was also considered in an earlier stage of this study. However, it turned out that this does not circumvent possible problems with missing or too little data. As exploratory analyses did not produce interpretable results, we decided to not follow this line of research and exclude it from the paper. Furthermore, we did not consider the segmentation according to Voronoi diagrams as an alternative option because it is unclear on the basis of which points they should be constructed.

Changes in manuscript: We added the following phrases in Section 3.1 of the paper (p. 4, l. 45): "This spatial segmentation was chosen because it makes the analysis compatible with other data (e.g. socio-economic data sets provided by the IBGE).

Additionally, the size of the municipalities reflect to some degree that of the population and therefore potential land-use activities. In principle, a segmentation into regular grid cells could provide complementary information and insights. However, to keep the presentation clear, we focus here on mesoregions and municipalities."

Referee comment: *2) Very minor technical points: – p.1, l. 14 world wide -> worldwide?*

Response and changes in manuscript: We changed the orthography as suggested from "world wide" to "worldwide".

Referee comment: *– p.11, section title "Results and Discussion", but there are many original results prior to this section, such as segmentation, network design etc. If not enforced by journal format, maybe use more specific (sub)sectioning?*

Response and changes in manuscript: We agree with the referee that the titles are very generic and changed the titles to more specific titles as follows:

2 Data: Land-cover maps of the Brazilian Amazon

3 A method to explore patterns of land-cover transitions

3.1 Extraction and normalization of transition matrices

3.2 Construction of similarity networks and clustering analysis of land-cover transitions

4 Spatial heterogeneity of land-cover transitions and discussion of clustering patterns

**3    Additional changes in the manuscript**

On the basis of the comment of referee 2 to make the subtitles more specific, we decided to also change the title of the paper to reflect the content of the paper in a better and clearer way. The new title reads: "A matrix clustering method to explore patterns of land-cover transitions in satellite-derived maps of the Brazilian Amazon"